# Histone Chaperones and Digestive Cancer: A Review of the Literature

**DOI:** 10.3390/cancers14225584

**Published:** 2022-11-14

**Authors:** Zhou Zhao, Zhaolun Cai, Tianxiang Jiang, Junhong Han, Bo Zhang

**Affiliations:** 1Research Laboratory of Tumor Epigenetics and Genomics, Department of General Surgery, West China Hospital, Sichuan University, Chengdu 610041, China; 2Division of Gastric Cancer Center, Department of General Surgery, West China Hospital, Sichuan University, Chengdu 610041, China

**Keywords:** histone chaperone, digestive cancer, prognosis, mechanism, review

## Abstract

**Simple Summary:**

Histone chaperones are responsible for histone metabolism. Recent investigations have revealed that the abnormal expression of several histone chaperones in digestive cancer is associated with the pathogenesis of digestive cancer. The purpose of this review is to determine the role of selected histone chaperones in the progression of digestive cancer and their prognostic significance. Selected histone chaperones are found to be associated with the proliferation, migration, chemosensitivity, and stemness of malignant cells, as well as a poor prognosis for patients with digestive cancer. In conclusion, this review confirms the significance of selected histone chaperones in the pathogenesis of the most common digestive cancer and highlights their potential as prognostic biomarkers and therapeutic targets for digestive cancer.

**Abstract:**

Background: The global burden of digestive cancer is expected to increase. Therefore, crucial for the prognosis of patients with these tumors is to identify early diagnostic markers or novel therapeutic targets. There is accumulating evidence connecting histone chaperones to the pathogenesis of digestive cancer. Histone chaperones are now broadly defined as a class of proteins that bind histones and regulate nucleosome assembly. Recent studies have demonstrated that multiple histone chaperones are aberrantly expressed and have distinct roles in digestive cancers. Objective: The purpose of this review is to present the current evidence regarding the role of histone chaperones in digestive cancer, particularly their mechanism in the development and progression of esophageal, gastric, liver, pancreatic, and colorectal cancers. In addition, the prognostic significance of particular histone chaperones in patients with digestive cancer is discussed. Methods: According to PRISMA guidelines, we searched the PubMed, Embase, and MEDLINE databases to identify studies on histone chaperones and digestive cancer from inception until June 2022. Results: A total of 104 studies involving 21 histone chaperones were retrieved. Conclusions: This review confirms the roles and mechanisms of selected histone chaperones in digestive cancer and suggests their significance as potential prognostic biomarkers and therapeutic targets. However, due to their non-specificity, more research on histone chaperones should be conducted in the future to elucidate novel strategies of histone chaperones for prognosis and treatment of digestive cancer.

## 1. Introduction

Digestive cancer refers to a highly heterogeneous group of malignant diseases, including esophageal cancer (EC), gastric cancer (GC), hepatocellular carcinoma (HCC), pancreatic ductal adenocarcinoma (PDAC), and colorectal cancer (CRC). According to the 2020 Global Cancer (GLOBOCAN) statistics [1], digestive cancer is the third leading cause of morbidity and mortality and has a significant impact on the global mortality burden. Each digestive cancer has different biological characteristics, such as cell proliferation, invasion, angiogenesis, treatment, and prognosis [2]. Even though the pathogenesis of these cancers has been studied extensively, much remains unclear. Therefore, further investigation of the specific genes involved in the development of digestive cancers may help to identify prognostic markers and new therapeutic targets.

Histone chaperones are a class of proteins responsible for histone metabolism, including histone storage, transport, post-translational modification, nucleosome assembly, and histone turnover [3]. Since the description of the first member of the histone chaperone family [4], more and more members have been identified [5]. Structural analysis of histone chaperone complexes contributes to the definition of histone chaperones, but there is less structural conservation among histone chaperones [3]. Individual histone chaperones vary widely in molecular structure, specificity, and spectrum of biological functions. Indeed, for some of them, their histone chaperone role was discovered after years of intensive research. The roles of many putative players are waiting to be confirmed [6]. Therefore, this paper only includes the 26 histone chaperones summarized by Hammond [3] (Appendix A).

There have been abundant studies on the relationships between histone chaperones and human cancers [7,8,9,10,11,12]. It has been reported that histone chaperones have a large potential prognostic value for multiple cancers, owing to their differential expression between tumor and normal tissues [8]. Meanwhile, upregulated or downregulated expression of these histone chaperones is related to an advanced tumor stage, tumor grade, metastasis, and overall survival [13]. Owing to the diverse structure and function of histone chaperones and the obvious heterogeneity of digestive cancer, the role of histone chaperones in each digestive cancer warrants further investigation. This review briefly outlines the recent developments in the function, mechanism, and potential clinical value of histone chaperones in each digestive cancer.

## 2. Literature Search

### 2.1. Search Process

See Appendix A for literature search details. PRISMA-S Checklist is presented in Appendix A. Figure 1 is a flow diagram illustrating the comprehensive study selection procedure.

### 2.2. Search Results

After selection, 104 eligible studies were included in this review (Appendix A). Only 20 of the 26 histone chaperones that we searched for have been thoroughly investigated in digestive cancers so far. In terms of tumor category, the majority of the included studies (33/104) focused on HCC, while EC received the fewest publications (5/104) (Figure 2A). In terms of phenotype, the most published studies are those related to proliferation (Figure 2B). In addition, almost all studies have explored the relationship between selected histone chaperones and proliferation (Figure 2C). Histone chaperones are involved in the histone metabolism, thus participating in the replication-independent and replication-coupled nucleosome assembly. This may explain why researchers prefer to investigate the connection between histone chaperones and proliferation. The annual number of studies is shown in Figure 2D. From 1996 to 2017, research on histone chaperones in digestive cancers grew slowly. However, more than ten articles were published each year after 2017. These data suggest that the role and mechanism of histone chaperones in digestive cancers are receiving increasing attention from researchers.

## 3. Histone Chaperones in Digestive Cancers 

### 3.1. The Histone Chaperone Network

The nucleosome core consists of an octamer containing two copies of the four core histones H3, H4, H2A, and H2B around which about 147 bp of DNA is wrapped. Linker DNA and linker histones H1 complete the nucleosome and connect neighboring nucleosomes [8]. Histone chaperones function as a network involved in the nuclear import, storage or buffering, folding or refolding, turnover, and degradation of histones. According to their selectivity for cognate histones, histone chaperones can be classified. Furthermore, since H3 exists as several variants, their specificity can be further refined. Previous studies identified Chromatin Assembly Factor-1 (CAF-1) complex with replicating H3.1 variants; Death Domain-Associated Protein (DAXX) and Histone Cell Cycle Regulator (HIRA) complex with the replacement variant H3.3; and Holliday junction recognition protein (HJURP) with human centromeric H3 variant centromere protein A (CENP-A). In contrast to these selective histone chaperones, the histone chaperones Anti-silencing Function 1 (ASF1) and Nuclear Autoantigenic Sperm Protein (NASP) are identified with both variant H3.1 and H3.3. In addition to these several histone H3 variant-specific histone chaperones, we summarize the histone chaperones for histones H1, H2A–H2B, H3–H4 in Figure 3.

### 3.2. Clinical Prognostic Value of Histone Chaperones in Digestive Cancer—General Information

Mounting evidence has associated an increased expression of individual histone chaperone with digestive cancers [8]. This review therefore focuses on their role in the pathogenesis of digestive cancers. The clinical significance of selected histone chaperones in digestive cancers is presented in Appendix A and Figure 4. The majority of histone chaperones play oncogenic roles in digestive cancers and correlate negatively with their prognosis. The roles of Nucleophosmin (NPM1), Nucleolin (NCL), DAXX, and Nucleosome Assembly Protein 1 Like 1 (NAP1L1) in certain digestive cancers remain controversial. Specific disputes are discussed in subsequent chapters.

### 3.3. Phenotype and Molecular Mechanisms of Histone Chaperones in Digestive Cancer—General Information

Recent studies indicated the importance of histone chaperones in tumor formation, migration, proliferation, and development [8]. Several studies confirmed the role of histone chaperones in digestive cancer cell biology. Characteristics of selected histone chaperones and their significance in tumor biology are presented in Table 1.

### 3.4. ASF1A/B in Digestive Cancer

Anti-silencing Function 1 (ASF1) is a conserved H3–H4 histone chaperone involved in replication-dependent and replication-independent chromatin assembly [87]. ASF1 encompasses two homologues, Anti-silencing Function 1A (ASF1A) and Anti-silencing Function 1B (ASF1B) [87]. Both ASF1A and ASF1B are involved in DNA replication, DNA damage repair, and gene transcription. ASF1A plays a crucial role in histone H3K56 acetylation and cellular reprogramming [88], whereas ASF1B is closely involved in proliferation regulation [89].

The vast majority of CRC (about 80%) displays Wnt pathway deregulation due to APC gene mutation [2]. APC mutation frees β-catenin to translocate to the nucleus and activate its target genes. Interestingly, Liang [13] revealed that ASF1A directly binds to β-catenin and promotes the activity of the β-catenin–ZEB1 axis, thereby promoting the proliferation, migration, and stemness of CRC cells. The discovery that ASF1A directly binds to β-catenin suggests a function beyond histone chaperone. An intriguing question is whether β-catenin is associated with ASF1A-mediated histone modification and histone assembly. Thus, this study provides some answers and raises deeper questions that will spur future research on the genetics-epigenetics link in cancer. Wu [14] discovered that knockdown of ASF1A induces DNA damage, resulting in growth arrest and senescence of HCC cells via activation of the p53/21^cip1^ axis. The function of ASF1A in cellular senescence and DNA damage repair has been the subject of numerous studies [90,91]. Previous research demonstrated that ASF1A and HIRA are required for the formation of senescence-associated heterochromatin [90]. Moreover, ASF1A-mediated H3K56 acetylation is required for nucleosome reassembly after DNA damage to facilitate DNA repair [91]. However, the findings of Wu suggest that DNA damage repair and cellular senescence mediated by ASF1A inhibition is p53-dependent. Wu did not elucidate why ASF1A inhibition leads to opposite results in cells with different p53 backgrounds. Therefore, this question is of great interest to cancer researchers, especially in tumors where P53 mutations are prevalent. In addition, a study revealed that ASF1A regulates tyrosine phosphorylation at Y72 of histone H4 (H4^Y72^) in CRC cells to promote autophagy [15]. The significance of autophagy in CRC is gaining recognition. However, autophagy has both anti-tumor and pro-tumor properties in colorectal cancer, depending on the context [92]. A previous study identified H4^Y72^ phosphorylation promotes DNA synthesis and repair [93]. Qiu believed that the ASF1A-H4^Y72^ phosphorylation axis promotes CRC autophagy via transcriptional regulation of ATG genes. However, the detailed mechanism needs to be studied in greater depth. This study offers a new perspective on autophagy and histone chaperones, which should be investigated in other chaperones.

ASF1B is overexpressed in HCC [16,18] and GC [19], and is negatively correlated with their prognoses. ASF1B is associated with multiple phenotypes of digestive cancer cells, including proliferation [16,19], migration [18], apoptosis [21], DNA damage [20], and chemosensitivity [20]. For upstream regulatory networks, Zhan reported that the miRNA-214-3p negatively transcriptionally regulates ASF1B [17]. However, Zhan did not delve into the function of miRNA-214-3p-ASF1B axis in HCC. The competing endogenous RNA (ceRNA) hypothesis proposes that mRNA and common noncoding RNAs can form complex regulatory networks and regulate protein expression post-transcriptionally by competitively binding to microRNA (miRNA) response elements [94]. A recent bioinformatics study [95] believed that circ_0002024 may function as a ceRNA in renal cancer via the miR_129-5p/ASF1B axis. The regulatory network map of long non-coding RNAs (lncRNAs), Circular RNAs (circRNAs), miRNAs, and histones requires elucidation and expansion in digestive cancers. In addition to miRNAs, transcription factors also play a significant role in the regulatory network of ASF1B. Hayashi [96] reported ASF1B as a direct transcriptional target of the transcription factor E2F1. However, the role and mechanism of this regulatory axis in digestive cancers have not been thoroughly studied. To comprehend the mechanism underlying the elevated expression of ASF1B in digestive cancers, it is essential to investigate its upstream transcription factors. For downstream events, Ouyang revealed that ASF1B interacts with CDK9 in HCC cells and induces cell cycle arrest [18]. Ouyang believed that ASF1B stabilizes CDK9 protein by inhibiting its proteasome-mediated ubiquitination and degradation. In addition, both Wang and Chen discovered that ASF1B affects PI3K-AKT and cell proliferation [19,21]. However, neither study elucidated the detailed mechanism by which ASF1B regulates the PI3K/AKT pathway.

The above studies show that ASF1A and ASF1B play a significant role in promoting cancer in the occurrence and development of digestive cancer. Accordingly, ASF1A was reported to be associated with OS and RFS in HCC and CRC [13,14]. However, two studies required multivariate analysis to determine whether ASF1A overexpression is an independent predictor of poor outcome. ASF1B is an independent poor prognostic factor for HCC and GC [18,19]. Additionally, the in vivo tumor-promoting roles of ASF1A and ASF1B were validated via cell line-derived xenograft (CDX) models of CRC and GC, respectively. Importantly, developed drugs that target ASF1A/B have demonstrated antitumor effects in vivo and in vitro [87,97,98]. Therefore, additional research is required to determine the role of ASF1A/B in various digestive cancers and the safety and effectiveness of its inhibitors.

### 3.5. FACT Complex in Digestive Cancer

In humans, the histone chaperone Facilitates Chromatin Transcription (FACT) complex is a heterodimeric protein complex consisting of SPT16 Homolog of Facilitates Chromatin Remodeling Subunit (SUPT16H, also known as SPT16) and Structure-Specific Recognition Protein 1 (SSRP1). It was initially described to loosen up the nucleosomes along the DNA strand to facilitate RNA polymerase II-driven transcription and then deposit back the histone proteins [99]. Furthermore, the FACT complex has been reported to be involved in DNA replication, DNA damage, transcription initiation, and transcription elongation [100].

Five studies have found SSRP1 to be overexpressed in digestive cancer, but only one study has reported SPT16 to be overexpressed [12,22,25,26,28]. In *Gut*, an elegant experimental and clinical study by Shen [12] elucidates the close and targetable relationships existing between the KEAP1/NRF2 pathway and the FACT complex in HCC cells. Shen [12] revealed that oxidative stress promotes the expression and stabilization of FACT complex via NRF2/KEAP1 pathway, while FACT complex promotes the transcription elongation of NRF2 and its downstream antioxidant genes via its histone chaperone functions. This study elegantly reveals the role of FACT in oxidative stress and sorafenib resistance via functions experiments in vitro, multiple subcutaneous and metastatic CDX mouse models, clinical studies, and comprehensive mechanistic studies. The KEAP1/NRF2 pathway is the key pathway providing defense against oxidative stress [101]. KEAP1/NRF2 is one of the most frequently mutated pathways in human HCC and associated with drug resistance, including sorafenib resistance [101]. Thus, this study elegantly elucidates the role and mechanism of FACT in HCC. Moreover, this study may guide curaxin (a FACT inhibitor) application in clinical research. For ceRNA network related to SSRP1, five studies have indicated that SSRP1 is negatively regulated by miR-497 [22,23], miR-4784 [24], miR-28-5p [28], and miR-584-3p [29] in HCC and CRC cells, respectively. Ding [22] believed that aberrant DNA gene methylation of miR-497 reduces miR-497 expression, resulting in increased SSRP1 levels in HCC. Zheng et al. [23,24,29] revealed that LINC01134, DLG1-AS1, and LOC101927746 function as ceRNAs for miR-4784, miR-497-5p, and miR-584-3p, respectively. These findings partially explain the abnormal expression of SSRP1 in digestive cancer. In addition to the post-translational modification of the ceRNA network, the molecular mechanisms by which SSRP1 promotes the malignant biological behavior of digestive cancers, such as transcriptional regulation and post-translational modification, remain to be further elucidated.

For downstream events, SSRP1 can regulate the PI3K-AKT pathway [25,27] and the Epithelial–mesenchymal transition (EMT) pathway [28], thereby regulating tumor cell proliferation and migration. Regretfully, none of the studies clarified how SSRP1 regulates the AKT and EMT pathways. Since numerous factors influence the AKT pathway, it remains debatable whether SSRP1 directly influences the AKT pathway. In a study on osteoblast differentiation [102], SSRP1 can affect β-catenin nuclear localization, that may affect the EMT pathway. Therefore, functions of SSRP1 beyond histone chaperones should be investigated, as ASF1A can directly bind β-catenin to stabilize it.

Unsurprisingly, SSRP1/SPT16 has also been reported to be negatively associated with the prognosis of HCC and CRC [12,22,28]. Notably, a multivariable analysis is required to determine if SSRP1/SPT16 overexpression is an independent predictor of poor outcome. In addition, the correlation between SSRP1/SPT16 and cancer prognosis in GC, EC, and PDAC requires further investigation.

### 3.6. HJURP in Digestive Cancer

HJURP plays a central role in the incorporation and maintenance CENP-A during the early G1 phase [103]. HJURP is identified as a crucial DNA-binding and phosphorylation factor that promotes chromosome segregation and cell mitosis in mammals [104]. In digestive cancer, HJURP has been reported to be highly expressed in HCC [9,30,31,32], PDAC [33], and CRC [34]. 

Hu and Li revealed that HJURP promotes cell proliferation and migration in vitro and is associated with poor clinical outcomes in HCC [31,32]. The two researchers did not, however, discuss the mechanism behind this phenomenon. Chen observed that HJURP can affect p21 ubiquitination by regulating the MAPK/ERK1/2 and AKT/GSK signaling pathways [9]. Regretfully, no report confirms the detailed connection between HJURP–CENP-A and the MAPK/ERK1/2 or AKT/GSK signaling pathways. Chen also revealed that HJURP can regulate SPHK1 to affect the EMT pathway, thereby affecting the migration of HCC cells [30]. Chen discovered an interaction between SPHK1 and HJURP via co-IP but was unable to establish its precise regulatory mechanism. Further investigation is required to determine whether HJURP functions beyond histone chaperones to promote the stability of other proteins. H3K4me2 is required for the targeting of HJURP and the assembly of CENP-A on a synthetic human kinetochore [105]. In addition, Wang reported that HJURP also regulates the MDM2/p53 axis via H3K4me2 and mediates cancerous behavior in PDAC [33]. In another recent study, CENP-A overexpression favored EMT pathway when p53 was inactive [106]. Therefore, whether CENP-A is involved in the HJURP-regulated EMT pathway and MDM2/p53 axis needs to be further confirmed.

Numerous studies have demonstrated that HJURP contributes to the development of cancer in digestive cancers. The HJURP expression is negatively associated with the overall survival of HCC [9,31,107] and CRC [34]. Other prognostic indicators, such as RFS, should also be investigated for their relationship to HJURP. In addition, HJURP was associated with HCC and PDAC proliferation in vivo via subcutaneous CDX models [9,33]. Nevertheless, in vivo validation of HJURP-mediated EMT pathway using a metastatic mouse model has not been performed. HJURP is currently studied infrequently in GC and EC. An inhibitor of HJURP has not yet been reported. HJURP has been demonstrated to play a significant role in numerous tumors, including prostate [108], breast [109], and ovarian cancers [110]. Therefore, it is essential to further investigate the function and mechanism of HJURP in tumors of the digestive tract.

### 3.7. CHAF1A/B in Digestive Cancer

CAF-1 was first identified as a factor facilitating replication-dependent chromatin assembly in vitro, consisting of Chromatin Assembly Factor 1 Subunit A (CHAF1A), Subunit A (CHAF1B), and p48 subunits in human [111]. CAF-1 functions as a histone chaperone that assembles newly synthesized histone H3/H4 onto replicating DNA [112]. Therefore, CAF-1 is essential for DNA replication during the process of nucleosome formation, as well as chromatin recovery after DNA repair [113].

Zheng and Xu [54,55] unearthed that CHAF1A expression is elevated in patients with HCC and GC and correlates with tumor cell proliferation. Mechanistically, Zheng [55] observed that *Helicobacter pylori* can induce CHAF1A expression in an SP1-dependent manner. The association between *Helicobacter pylori* infection and CHAF1A expression provides partial insight into GC carcinogenesis and the reason CHAF1A is highly expressed in GC. Furthermore, Zheng [55] also revealed that CHAF1A interacts directly with TCF4 to transcriptionally enhance the expression of c-MYC and CCND1 in GC. The ability of CHAF1A to bind TCF4 is beyond the function of traditional histone chaperones. Similarly, it has been reported that ASF1A performs the same function. ASF1A can bind β-catenin to promote EMT pathway in GC and CRC [13]. CAF-1 and ASF1 cooperatively participate in replication-coupled nucleosome assembly via interaction with histone H3/H4 [8]. Consequently, it is worthwhile to investigate whether ASF1A/β-catenin/TCF4/CHAF1A forms a complex that contributes to the EMT pathway. In addition, another issue to investigate is whether CHAF1B, subunit of CAF-1, is involved. Wang also reported that CHAF1A is associated with thymidylate synthetase, a chemosensitizing effector [56]. Regrettably, the authors hypothesized a mechanism via bioinformatics without experimental validation. For clinical prognostic value of CHAF1A, Xu and Wang reported that CHAF1A expression is associated with unfavorable prognosis in HCC and GC [54,56]. However, multivariate analysis to determine whether CHAF1A is an independent prognostic factor can better explain its prognostic role. For CHAF1B, there is only one report indicating that knockdown of CHAF1B hinders the proliferation and migration phenotype of HCC cells [57]. However, the researchers did not explain how CHAF1B affects HCC biology. In vivo studies revealed the pro-tumorigenic effect of CHAF1A/B, showcasing that its overexpression promoted tumor growth [54,55,57]. However, no established mouse model of metastasis exists to confirm the role and mechanism of CHAF1A/B in digestive cancers metastasis in vivo.

More recently, CHAF1A has been implicated in the development and progression of solid tumors, including prostate cancer [114], glioma [115], and neuroblastoma [116]. Nevertheless, research on its role in digestive cancer is still uncommon. Future research is required to clarify its function and mechanism in digestive cancers.

### 3.8. NPM1 in Digestive Cancer

NPM1 is a member of the NPM family (NPM1–3) with the most research. Multifunctional NPM1 is a nucleocytoplasmic shuttling phosphorylated protein, involved in multiple cellular processes, such as histone assembly, mRNA transport, ribosome biogenesis, centrosome duplication, nuclear localization, and genome stability [117]. NPM1 also acts as a histone chaperone for the histones H2B, H3, and H4 [117]. In addition, NPM1 gene mutations occur frequently in numerous tumors, particularly in hematological cancers [118]. Despite the lack of evidence for NPM1 mutations in digestive cancer, numerous studies have demonstrated the aberrant expression and significant role of NPM1 in digestive cancer, including HCC [39,119], CRC [44,45,46,47,120,121], GC [122,123,124], and PDAC [41,42]. In the included literature, NPM1 ranks among the highest in terms of the number of articles. However, NPM1 appears to function independently of its histone chaperone function in digestive cancers.

First, the transcriptional regulatory function of NPM1 promotes the progression of digestive cancers. Extensive research has been conducted on the function of NPM1 as a crucial cofactor in c-Myc transforming activity [117]. In EC and CRC, the NPM1/c-MYC axis promotes cancerous behavior [46,49]. NPM1 also acts as a corepressor or coactivator of transcription by binding to transcription factors. Tang found that NPM1 regulates CRC progression by activating the NF-κB pathway via its interaction with p65 [43]. Wang observed that CBX3 and NPM1 promote PRDX6 transcription as a complex in CRC [48]. In PDAC, Zhu reported that NPM1 represses FBP1 transcription to promote tumorigenicity by binding to its promoter [41]. However, Zhu did not elaborate on the function of NPM1 in promoting transcription. In general, NPM1 regulates RNA polymerase I/II-driven transcription, acetylates core histones, and functions as a co-repressor or co-activator [117]. Histone chaperone functions of NPM1 are presumed to be essential for transcription, but the detailed mechanism is mysterious.

Second, as a nucleocytoplasmic shuttle protein, NPM1 plays a crucial role in protein modification, synthesis, degradation, and localization [117]. Luo discovered that the cytoplasmic translocation of NPM1 to regulate BAX function contributes to HCC cell death evasion [39]. Liu revealed that NPM1-promoted ATF5 degradation antagonizes the inhibitory effect of ATF5 on HCC cell proliferation [36]. Under hypoxic conditions in HCC cells, NPM1 binds to PTPN14 and prevents it from interacting with YAP via retaining PTPN14 in the nucleus [35]. Moreover, NPM1 can affect the PI3K-AKT pathway in CRC cells, thereby amending chemosensitivity [45,46]. The authors hypothesized that NPM1 may form complexes with PIP3 and AKT but did not confirm this association. According to a study published in *PNAS*, AKT interacts directly with NPM1 and protects it from apoptotic cleavage [125]. According to Hamilton, AKT phosphorylates NPM1 on serine 48 to regulate the localization of ARF in PDAC cells [42]. The interaction between ARF and NPM1 has been intensively investigated [117]. ARF can affect the nucleocytoplasmic shuttling and stability of NPM1. In turn, NPM1 regulates the localization and stability of ARFs in the nucleolus. Therefore, further research is required to determine the function and mechanism of the AKT/NPM1/ARF regulatory network in digestive cancers.

Third, NPM1 protein is extensively modified by phosphorylation, acetylation, ubiquitination, and SUMOylation [117]. These modifications affect the localization, oligomerization, and stability of NPM1. Ching reported that CDK1-mediated phosphorylation of NPM-Thr^234/237^ is crucial for HCC metastasis via activation of the Rho kinase II signaling pathway [38]. In publications outside of digestive cancers, NPM1 is also phosphorylated by casein kinase 2, polo-like kinase 2, CDC2, and cyclin E/CDK2 complex [117]. The role of these factors on NPM1 in digestive cancers remains to be investigated.

NPM1 is currently believed to be a cancer promoter in HCC, CRC, and PDAC. In HCC, NPM1 is associated with poor OS and RFS [39,119,126]. In CRC, NPM1 overexpression is prevalent [44,46,120] and negatively correlates with its prognosis [45,47,121]. However, the clinical value of NPM1 in GC appears controversial. Three studies reported high NPM1 expression in GC and its association with poor prognosis [122,123,124], while one reported the opposite results [127]. NPM1 is a nucleocytoplasmic shuttle protein. Different roles for NPM1 may exist in the nucleus and cytoplasm of tumor cells. Consequently, differences in nuclear/cytoplasmic expression of NPM1 should further establish its clinical prognostic value.

NPM1 is considered as a “storage platform” or “sink” due to its ability to store histones for a prolonged time before the actual transfer [117]. NPM1’s histone chaperone functions are presumed to be essential at each cellular location. However, the detailed mechanism of NPM1 in transcription, replication, and repair remains obscure. Interestingly, enforced NPM1 expression increases cellular resistance to UV-induced cell death [117]. Therefore, NPM1 can potentially function as a histone chaperone during DNA strand break repair of tumor cells. Furthermore, NSC348884, an NPM1 inhibitor that disrupts oligomer formation and induces apoptosis, has demonstrated antitumor activity in CRC and Ewing sarcoma cells in vitro [45,128]. Further research is required on NPM1 inhibitors in digestive cancers.

### 3.9. NCL in Digestive Cancer

NCL is a protein with multiple functions, including ribosome biosynthesis, rRNA transcription, nucleosome remodeling, translation regulation, and angiogenesis [129]. The distribution of NCL is ubiquitous, mainly in the nucleolus. Angelov [130] reported for the first time in 2006 that NCL serves as a histone chaperone. They revealed that NCL binds directly to H2A–H2B dimers and promotes their assembly into nucleosomes on naked DNA [130]. NCL acts like FACT, promoting chromatin transcription elongation by promoting the removal of H2A–H2B dimers during transcription [129]. However, the function of NCL as a histone chaperone has been poorly illuminated by studies of digestive cancers.

NCL on the cell surface functions as a low-affinity receptor for particular ligands. Interestingly, NCL is highly expressed in diverse tumor cells and specifically localized to the cell surface [129]. The majority of HCC and EC tissues displayed NCL staining at the cell surface [58,131]. Qiu [58] reported that cell–surface NCL is associated with an advanced stage, lymph node metastasis, and a poor 5-year prognosis. Further research revealed that cell–surface NCL is essential for the initiation of the CCR6 pathway [58]. Chen revealed that cell–surface NCL is an HDGF receptor, that mediates HDGF-stimulated oncogenic behavior and PI3K/Akt pathway in HCC [59]. Interestingly, the VEGF-mediated PI3K/Akt pathway promotes the translocation of NCL to the cell surface [64]. In other tumors, cell–surface NCL binds to a variety of angiogenesis-related ligands, such as endostatin, tumor-homing peptide F3, and P-selectin [129]. NCL, as a shuttle protein, seems to act as an intermediary in networks of multiple signaling pathways.

Five studies reported that NCL is highly expressed in HCC [59,132], PDAC [62], GC [61], and CRC [64]. In addition, high NCL expression is an independent indicator of poor prognosis. However, these researchers did not conduct an in-depth analysis according to the diverse localizations of NCL. Nuclear NCL is associated with rRNA synthesis, ribosome assembly, nucleosome assembly, RNA polymerase I transcriptional regulation [129]. Surprisingly, however, nuclear NCL appears to be a protective factor for digestive cancer prognosis [133]. According to Qiu and Peng [133,134], nuclear and global NCL are highly expressed in GC and PDAC compared to non-malignant tissues. However, high expression of nuclear NCL is associated with better prognosis. We might speculate that the high expression of nuclear NCL may be an effect of tumor progression, while the high extra-nuclear expression may be the cause. However, the detailed mechanism of NCL translocation remains obscure. The answer to this question will reveal why NCL is highly expressed on the surface of certain tumor cells.

High expression of cell-surface NCL is a frequent event in digestive cancers. Consequently, several NCL inhibitors have been invented and shown to be effective in preclinical studies. Gilles [62] found that N6L, a synthetic pseudopeptide that selectively binds to NCL [135], decreases the secretion of angiopoietin 2 and induces normalization of tumor blood vessels. Raineri [136] also observed that N6L hinders the growth of PDAC by inhibiting the activation of the Wnt/β-catenin pathway. LZ1, another NCL inhibitor, has been shown to induce autophagy-dependent cell death in PDAC by targeting the NCL/AMPK/autophagy axis [63]. Other NCL inhibitors, such as the diterpene oridonin and AS1411, have been demonstrated to possess antitumor activity in a variety of tumors [137,138,139]. Importantly, N6L has completed the Phase I/IIa study (NCT01711398), but no research data have been published.

### 3.10. NAP1L1 in Digestive Cancer

Nucleosome Assembly Protein 1 (NAP1) was first recognized as a histone chaperone associated with histone H2A–H2B dimerization and a factor promoting in vitro nucleosome assembly [140]. NAP1 participates in DNA replication, transcriptional regulation, and DNA repair, thereby aiding in the regulation of cell proliferation [140]. In humans, there are four members of the NAP1 family (NAP1L1 to NAP1L4). NAP1L1 is the most researched family member in HCC and CRC.

Chen and Huang revealed that NAP1L1 can stimulate the PI3K/AKT pathway to facilitate the progression of HCC [11,52]. Similar to other histone chaperones, the detailed mechanism underlying NAP1L1’s effect on the PI3K/AKT pathway is unidentified. Similar to NPM1, CHAF1A, and ASF1A, NAP1L1 also binds to transcription factors and functions as a coactivator. Zhang [50] demonstrated that NAP1L1 interacts with c-Jun and activates the c-Jun/CCND1 axis to promote the tumorigenicity of HCC cells. Furthermore, NAP1L1 is associated with the methylation of gene promoter. Schimmack [53] reported that NAP1L1 directly interacts with the p57^Kip2^ promoter and inhibits its transcription. NAP1L1 may bind p57^Kip2^ transcription factors such as HDACI/II to perform this function, but the authors did not experimentally confirm it. In short, current research on NAP1L1 in digestive cancers has focused on its non-histone chaperone function.

There are two published explanations for the high expression of NAP1L1 in digestive cancers. Huang [52] revealed that NAP1L1 is a direct target of let-7c-5p, a tumor suppressor miRNA in HCC. Chen demonstrated that PRDM8 could bind to NAP1L1 and prevent it from binding downstream molecules in HCC cells [11]. Given the rarity of NAP1L1 mutations in digestive cancers, its upstream regulatory network, such as the ceRNA network, warrants investigation.

The clinical significance of NAP1L1 as a cancer promoter in digestive cancers is suggested. Zhang and Le [50,51] reported that NAP1L1 is highly expressed in HCC, indicating an unfavorable overall survival. In CRC, Queiroz [141] also demonstrated the elevated expression and prognostic significance of NAP1L1. NAP1L1 also exhibits abnormal expression levels and significant prognostic value in other tumors, such as breast cancer [142], glioma [143], and ovarian cancer [144].

### 3.11. DAXX and ATRX in Digestive Cancer

DAXX was initially identified as a FAS-binding protein and a JNK-mediated cell death regulator [145]. ATRX gene mutations are known to cause alpha-thalassemia, mental retardation, and X-linked (ATRX) syndrome [146]. The ATRX/DAXX chromatin remodeling complex catalyzes the replication-independent deposition of histone H3.3 in central DNA repeats and telomeres [147]. As transcriptional factor coregulator involved in cell proliferation and apoptosis, the role of ATRX/DAXX in cancer development is controversial [148]. The first tumor to observe ATRX or DAXX loss mutations is pancreatic neuroendocrine tumors [149]. Notably, the loss of ATRX/DAXX expression is an independent prognostic biomarker for decreased RFS in non-functional pancreatic neuroendocrine tumors [150]. In addition, ATRX loss mutations are also prevalent in low-grade glioma, glioblastoma, neuroblastoma, and osteosarcoma [151]. In neuroendocrine tumors and breast cancer, DAXX functions as a tumor suppressor [148]. However, in prostate cancer, glioblastoma, and ovarian cancer, DAXX expression is upregulated and promotes tumor progression and chemoresistance [148]. Therefore, ATRX and DAXX may promote or inhibit malignant neoplasms, depending on the cancer type.

In digestive cancer, ATRX and DAXX are rarely mutated, but their expression and prognostic value also play paradoxical roles. In GC, Wu [68] reported that DAXX inhibits cancer stemness and EMT. However, Chen and Xu [67,152] found that DAXX function depends on subcellular localization. Chen [67] demonstrated that decreasing DAXX expression in the nucleus inhibits proliferation, migration, and invasion of GC cells, while elevated DAXX expression in the cytoplasm promotes migration and invasion. Based on the reported literature [69,70,72], the function of DAXX in CRC also appears to be contradictory. Chen [69] discovered that DAXX is expressed at a low level in CRC tissues and inhibits proliferation and migration. However, Huang [71] reported that DAXX is highly expressed in CRC tissues and stimulates proliferation in vivo and in vitro. The only publication on the role of DAXX in EC indicated that DAXX is highly expressed in EC and is associated with poor prognosis [153].

The role of DAXX in transcriptional regulation is well documented. DAXX functions as a transcriptional coregulator via interactions with a growing number of transcription factors and other nuclear proteins [148]. Liu and Tzeng [70,72] demonstrated that DAXX can function as a transcription corepressor by binding to ZEB1 and TCF4 to inhibit the EMT pathway of CRC cells. Another study discovered that DAXX recruits HDAC-1 to inhibit SNAI3 transcription of GC cell [68]. Therefore, they observed that DAXX inhibited tumor migration via EMT pathways. DAXX also has a role in transcriptional activation in digestive cancers. Huang [71] reported that DAXX binds to a potential transcription factor in the PIK3CA promoter region and stimulates its transcription in CRC cells. Daxx silencing significantly reduced CRC cell growth in vivo and in vitro. Huang speculated via bioinformatics that the transcription factor may be NF-κB but did not confirm it experimentally [71]. Although the interaction between Daxx and NF-κB has been reported [154], further studies are still needed to elucidate the transcription factors required for DAXX to activate PIK3CA transcription in CRC cells. Intriguingly, the above research revealed contradictory functions of DAXX in GC and CRC. A study on the opposing biological functions of cytoplasmic and nuclear DAXX in GC may offer another perspective. Chen [67] observed that a high cytoplasmic DAXX was associated with a better prognosis, while a high nuclear DAXX expression suggested a poorer survival. Further research found that SUMO-2/3 directly binds to DAXX and regulates its subcellular localization and distribution. In conclusion, current research on DAXX in digestive tumors has focused on DAXX transcriptional regulation. The modification, intracellular distribution, and function of DAXX as cofactors all play significant roles in digestive cancers.

There are currently a limited number of studies investigating the relationship between ATRX and digestive cancer. In PDAC, loss of ATRX increases tumorigenicity in genetically engineered KRAS^G12D^ mouse [74]. When ATRX function is deficient, chromosomal integrity of cancer cells is at stake. In this respect, the maintenance of telomeres is particularly revealing and may partially explain the increased tumorigenicity caused by ATRX loss in PDAC. However, in CRC, Li [73] discovered that ATRX is positively regulated by JMJD1A and contributes to the development of CRC. Regretfully, the authors did not elaborate on the mechanism by which ATRX promotes the progression of CRC. However, this serves as a reminder that the current understanding of ATRX as a tumor suppressor in digestive cancers is incomplete, as ATRX may also act as a tumor promoter depending on the context.

### 3.12. Other Histone Chaperones in Digestive Cancer

In addition to the above-mentioned histone chaperones, the roles of other histones in digestive cancers are being researched. MCM2, a well-recognized proliferation marker and histone chaperone, is highly expressed in HCC, EC, GC, and CRC, and negatively correlates with their prognosis [155,156,157,158]. Interestingly, less is known about the regulatory network mechanism of MCM2 in digestive cancer. Ahmed and Wang [75,76] discovered that miR-34a-5p and miR-195-5p/497-5p negatively regulate MCM2 to mediate HCC proliferation and CRC stemness, respectively. In addition, it has been reported that TONSL, RBAP48, NASP, IPO4, SPT2, SPT6, ANP32E, and VPS72 also promote the progression of digestive cancer through a variety of mechanisms, although data regarding their association with prognosis are limited (Appendix A and Table 1).

### 3.13. Histone Chaperone-Targeted Cancer Therapeutics

To date, several histone chaperone inhibitors have been reported. Among them, Curaxin is regarded as the most promising drug. Curaxin is an inhibitor of the FACT complex by intercalating into DNA and causing indirect chromatin-trapping of this histone chaperone [159]. Curaxin not only inhibits tumor growth, but also enhances sensitivity to chemotherapy and radiation therapy [160,161]. Currently, three clinical trials of Curaxin for the treatment of tumors are ongoing or have been concluded (ClinicalTrials.gov Identifier: NCT03727789, NCT01905228, and NCT04870944). Therefore, the FACT complex is a potential therapeutic target for digestive cancer, and the role and mechanism of its inhibitors in digestive cancer should be investigated further.

Multiple teams have designed peptides that inhibit the ASF1A/B-histone interaction utilizing structural, computational, and biochemical techniques [87,97,98]. Importantly, Developed ASF1A/B inhibitors have demonstrated antitumor effects in vivo and in vitro. These findings therefore pave the way for the application of ASF1 inhibitors in cancer treatments. Regrettably, commercialization of these inhibitors has not yet occurred. Mbianda [87] stated in the publication of Science Advances that the application of new technologies may be able to conceive potent second-generation ASF1 inhibitor. Thus, ASF1 inhibitors show promise as histone chaperone inhibitors similar to Curaxin.

Additionally, NCL inhibitors and NCL inhibitors have been documented. However, their effects appear less connected to histone chaperone functions. NSC348884, an NPM1 inhibitor that disrupts oligomer formation and induces apoptosis, has demonstrated antitumor activity in CRC and Ewing sarcoma cells in vitro [45,128]. Further research is required on NPM1 inhibitors in digestive cancers. High expression of cell-surface NCL is a frequent event in digestive cancers. Consequently, several NCL inhibitors have been invented and shown to be effective in preclinical studies. Gilles [62] found that N6L, a synthetic pseudopeptide that selectively binds to NCL [135], decreases the secretion of angiopoietin 2 and induces normalization of tumor blood vessels. Raineri [136] also observed that N6L hinders the growth of PDAC by inhibiting the activation of the Wnt/β-catenin pathway. LZ1, another NCL inhibitor, has been shown to induce autophagy-dependent cell death in PDAC by targeting the NCL/AMPK/autophagy axis [63]. Other NCL inhibitors, such as the diterpene oridonin and AS1411, have been demonstrated to possess antitumor activity in a variety of tumors [137,138,139]. Importantly, N6L has completed the Phase I/IIa study (NCT01711398), but no research data have been published.

Recent studies provide a rationale for a novel combination therapy consisting of histone chaperone inhibition and immunotherapy. A study [162] published in *Nature* showed that Curaxin reverses immune checkpoint blockade unresponsiveness in a refractory cutaneous malignant melanoma model. Notably, Curaxin manifested the effect by activating ZBP1 in tumor microenvironment fibroblasts. Another study [163] discovered that knocking out ASF1A in Kras-mutated lung adenocarcinoma cells increases tumors sensitivity to checkpoint blockade. This study provides a rationale for a potential therapy combining ASF1A inhibition and anti-PD-1 immunotherapy, although the authors did not investigate ASF1 inhibitor. In the future, it will be crucial to figure out whether other histone chaperones inhibition can enhance the efficacy of immunotherapy.

## 4. Conclusions

Digestive cancer is one of the most prevalent cancers among men and women worldwide. Therefore, there is an urgent need for more research on diagnostic or prognostic markers and new therapeutic targets for these cancers. Histone chaperones are now broadly defined as a class of proteins that bind histones and regulate nucleosome assembly. In the present study, 26 histone chaperones were screened for research on the functions and mechanisms of digestive cancer. This study demonstrated that the majority of the histone chaperones promote the development of digestive cancer. However, some histone chaperones, such as nucleolin, ATRX, and DAXX, appear to play contradictory roles, requiring further investigation.

In conclusion, our review confirms that histone chaperones play a crucial role in the development and prognosis of digestive cancer. We identified several issues that need to be addressed in histone chaperones. First, the roles and mechanisms of some histone chaperones, such as RBAP48, IPO4, SPT2, and SPT6, in digestive cancer have only recently come to light. Second, it is worthwhile investigating the function of histone chaperones beyond histone chaperones. Some multifunctional histone chaperones, such as NPM1 and NCL, appear to function independently of histone chaperones in tumors. The role of DAXX in transcriptional regulation is well documented. Recent studies revealed that ASF1A, CHAF1B, and NAP1L1 appear to function as transcriptional cofactors in transcriptional regulation. Consequently, the histone chaperone-independent functions of these molecules may also play a crucial role in the progression of tumors. Third, investigating the upstream regulatory network of histone chaperones contributes to understanding the frequent high expression of these molecules in digestive cancers. In digestive cancers, mutations in histone chaperones are infrequent. Networks of transcription factors and microRNAs aid in the explanation of abnormal expression. Fourth, in vivo mouse models can provide insight into the function of histone chaperones. The majority of current studies employ subcutaneous CDX models, which can only investigate tumor cell proliferation. Metastatic or orthotopic CDX models can better mimic the primary tumor and metastases. Additionally, CDA or GEM mouse models can aid in elucidating the function of histone chaperones in the tumor immune microenvironment. Finally, inhibitors of histone chaperones require further investigation. Several histone chaperone inhibitors, such as ASF1 inhibitors, FACT inhibitors, and NPM1 inhibitors, have been developed. Their preclinical and clinical research on digestive cancers is valuable. Additionally, the efficacy of these inhibitors in combination with other drugs, such as chemotherapy or immunotherapy, warrants investigation.

## Figures and Tables

**Figure 1 cancers-14-05584-f001:**
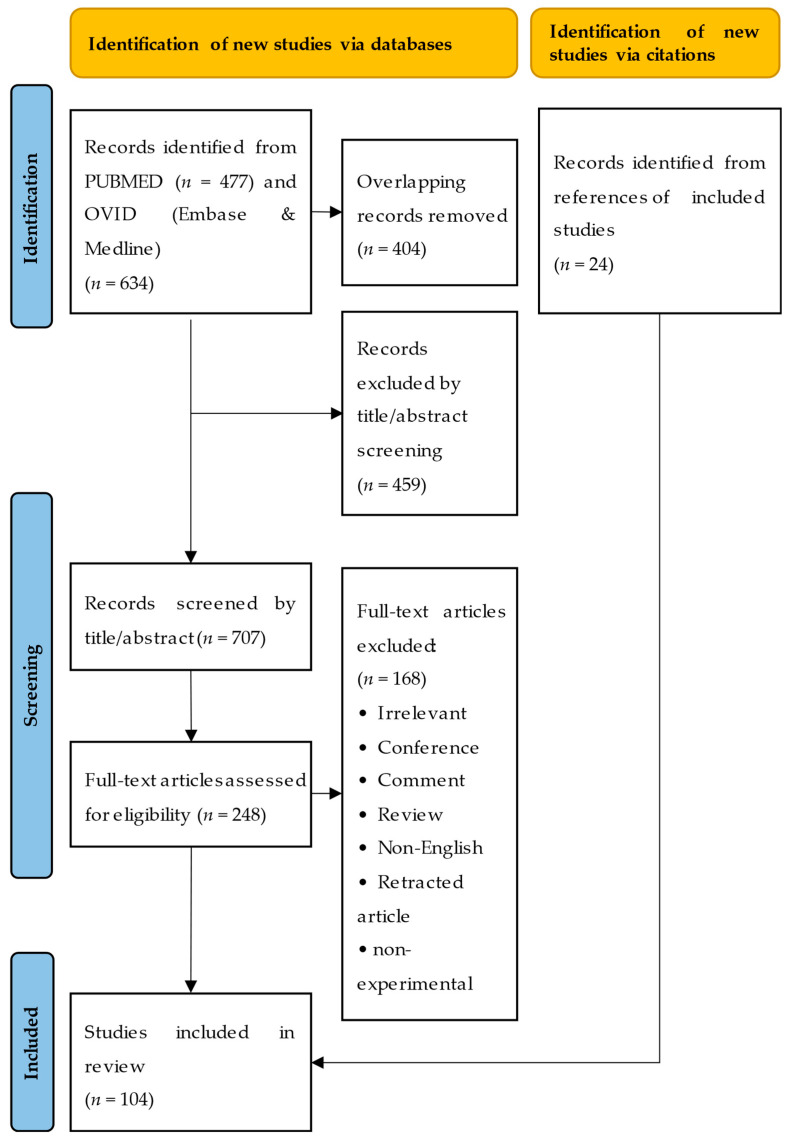
PRISMA flow diagram. Presentation of the procedure of literature searching and selection with numbers of articles at each stage.

**Figure 2 cancers-14-05584-f002:**
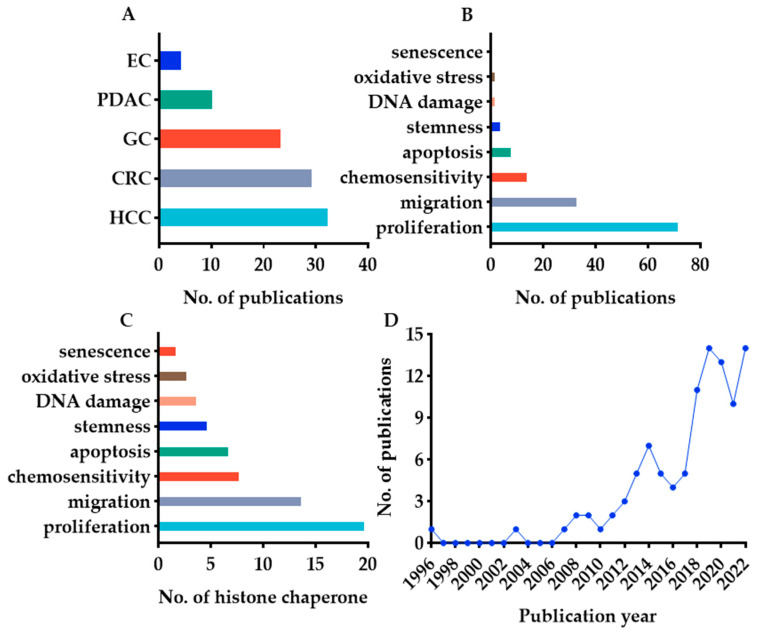
Search results of histone chaperones in digestive cancers. (**A**) Number of publications for selected histone chaperones in each digestive cancer; (**B**) number of publications for each phenotype of selected histone chaperones; (**C**) number of histone chaperones involved in each phenotype; (**D**) annual publications of histone chaperones in digestive cancers.

**Figure 3 cancers-14-05584-f003:**
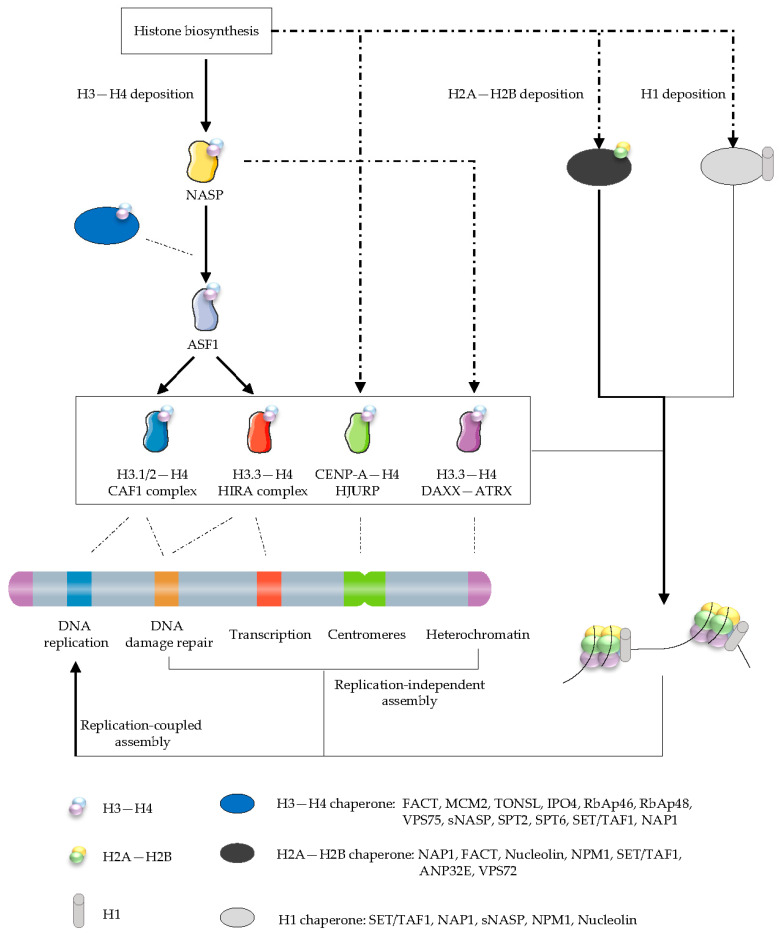
The histone chaperone network for histone supply. Newly synthesized histones H3–H4 and H2A–H2B interact with multiple chaperones and enzymes on their way to chromatin after synthesis. ASF1 and NASP are central chaperones in delivery of H3.1/H3.2– and H3.3–H4 dimers. The H3.1/H3.2– and H3.3–H4 dimers are shuttled by ASF1 to the CAF1 and HIRA complexes for replication-coupled and replication-independent assembly, respectively. In addition, DAXX-ATRX also deposits H3.3–H4 dimers, while HJURP deposits CENP-A–H4 dimers. Histone chaperones such as NAP1 and FACT complex handle H2A–H2B dimers. Linker histones H1 have been shown to interact with histones chaperones such as SET/TAF1, NASP, and NAP1 in the nucleus and cytoplasm, but the chronological sequence of these interactions is still mysterious. Definitions and classifications of histone chaperones are shown in Appendix A.

**Figure 4 cancers-14-05584-f004:**
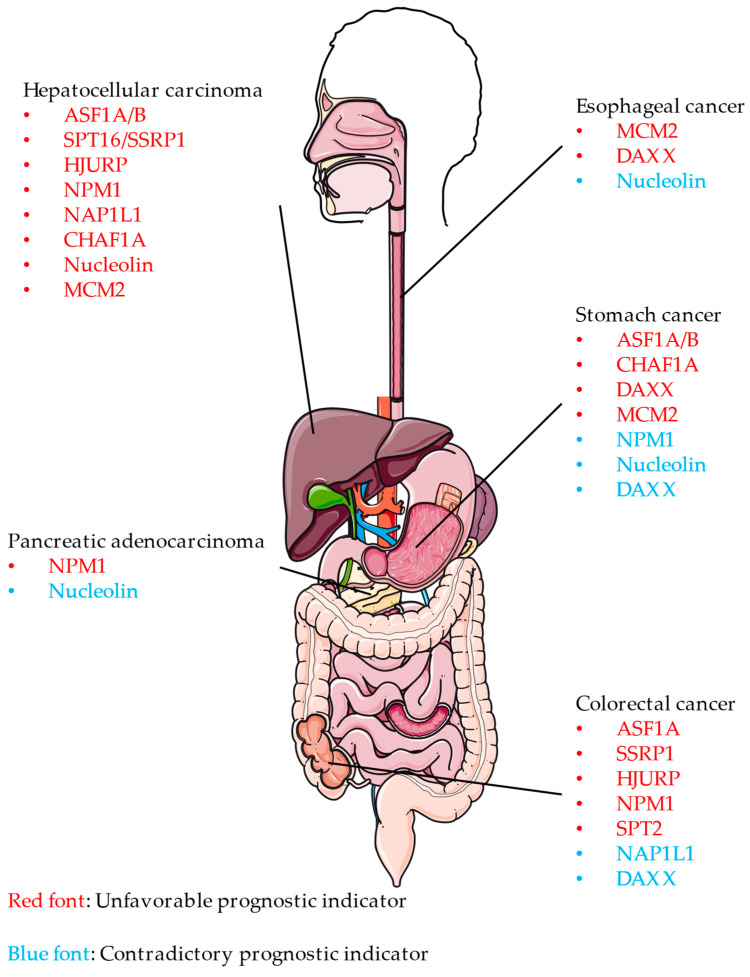
The significance of selected histone chaperones in digestive cancers. Image was obtained from Smart Servier Medical Art (https://smart.servier.com; accessed on 19 October 2022).

**Table 1 cancers-14-05584-t001:** Phenotype and mechanism of selected histone chaperones in digestive cancer.

Gene	Tumor	Role	Effect In Vitro	Effect In Vivo (Mouse Model)	Upstream Events	Downstream Events	Reference Number
ASF1A	HCC	promoter	cell senescence, DNA damage	NR	NR	p53/p21^cip1^ pathway	[14]
	CRC	promoter	proliferation, migration, stemness	proliferation and migration (metastatic and subcutaneous CDX model)	NR	β-catenin	[13]
	CRC	promoter	NR	NR	NR	H4^Y72ph^/autophagy	[15]
ASF1B	HCC	promoter	proliferation	NR	NR	NR	[16]
	HCC	promoter	NR	NR	miRNA-214-3p	NR	[17]
	HCC	promoter	proliferation, migration	NR	NR	CDK9 stabilization	[18]
	GC	promoter	proliferation migration	proliferation (subcutaneous CDX model)	NR	PI3K/AKT/mTOR pathway	[19]
	PDAC	promoter	proliferation, migration, DNA damage, chemosensitivity	NR	NR	NR	[20]
	PDAC	promoter	proliferation, apoptosis	NR	NR	PI3K/AKT pathway	[21]
SPT16	HCC	promoter	proliferation, migration, oxidative stress chemosensitivity	proliferation (subcutaneous and orthotopic CDX model)	NRF2/KEAP1 pathway	NRF2 transcription elongation rate	[12]
SSRP1	HCC	promoter	proliferation, migration, oxidative stress chemosensitivity	proliferation (subcutaneous and orthotopic CDX model)	NRF2/KEAP1 pathway	NRF2 transcription elongation rate	[12]
	HCC	Promoter	proliferation, migration, apoptosis, chemosensitivity	proliferation and migration (metastatic and subcutaneous CDX model)	miR-497	NR	[22]
	HCC	NR	NR	NR	DLG-AS1/miR-497-5p	NR	[23]
	HCC	promoter	proliferation	NR	LNC01134/miR-4784	NR	[24]
	GC	promoter	proliferation, migration, apoptosis	NR	NR	AKT pathway	[25]
	CRC	promoter	DNA damage, chemosensitivity	chemosensitivity (subcutaneous CDX model)	NR	APE1 acetylation	[26]
	CRC	promoter	proliferation, migration	proliferation (subcutaneous CDX model)	NR	AKT pathway	[27]
	CRC	promoter	proliferation, migration, chemosensitivity	proliferation (subcutaneous CDX model)	miR-28-5p	MMP9	[28]
	CRC	promoter	NR	NR	LOC101927746/miR-584-3p	NR	[29]
HJURP	HCC	promoter	proliferation	proliferation (subcutaneous CDX model)	NR	MAPK/ERK1/2 and AKT/GSK3β pathways	[9]
	HCC	promoter	migration	NR	NR	SPHK1	[30]
	HCC	promoter	proliferation	NR	NR	NR	[31]
	HCC	promoter	proliferation, migration	NR	NR	NR	[32]
	PDAC	promoter	proliferation, migration	proliferation and migration (subcutaneous CDX model)	NR	H3K4me2/MDM2/p53 axis	[33]
	CRC	promoter	proliferation, migration	NR	NR	NR	[34]
NPM1	HCC	promoter	proliferation, chemosensitivity	NR	NR	PTPN14/YAP axis	[35]
	HCC	promoter	proliferation	NR	NR	ATF5 degradation	[36]
	HCC	promoter	proliferation, migration	NR	NR	NR	[37]
	HCC	promoter	migration	NR	CDK1	Rho/ROCK/LIMK pathway	[38]
	HCC	promoter	chemosensitivity	NR	NR	BAX mitochondria translocation and oligomerization	[39]
	GC	NR	NR	NR	RASSF10	RNF2/RASSF10 feedback	[40]
	PDAC	promoter	proliferation	NR	NR	FBP1	[41]
	PDAC	promoter	NR	NR	AKT	ARF localization	[42]
	CRC	promoter	proliferation	NR	DDX27	NF-κB pathway	[43]
	CRC	promoter	proliferation, senescence	NR	NR	NR	[44]
	CRC	promoter	proliferation, chemosensitivity	proliferation (subcutaneous PDX model)	NA	PI3K/AKT pathway	[45]
	CRC	promoter	chemosensitivity	NR	NR	c-MYC	[46]
	CRC	promoter	proliferation, migration	NR	NR	NR	[47]
	CRC	promoter	proliferation, oxidative stress	proliferation (subcutaneous CDX model)	NR	CBX3/PRDX6 axis	[48]
	EC	NR	NR	NR	SLC25A21-AS1	c-MYC	[49]
NAP1L1	HCC	promoter	proliferation	proliferation (subcutaneous CDX model)	NR	HDGF/c-Jun/CCND1 axis	[50]
	HCC	promoter	proliferation	NR	PRDM8	PI3K/AKT/mTOR pathway	[11]
	HCC	promoter	proliferation, chemosensitivity	proliferation and chemosensitivity (subcutaneous CDX model)	NR	NR	[51]
	HCC	promoter	proliferation	NR	Let-7c-5p	PI3K/AKT/mTOR pathway	[52]
	PDAC	promoter	proliferation	proliferation (orthotopic CDX model)	NR	NR	[53]
CHAF1A	HCC	promoter	proliferation, apoptosis,	proliferation (subcutaneous CDX model)	NR	NR	[54]
	GC	promoter	proliferation	proliferation (subcutaneous CDX model)	SP1	TCF4/c-MYC/CCND1 axis	[55]
	GC	promoter	chemosensitivity	NR	NR	Thymidylate synthetase	[56]
CHAF1B	HCC	promoter	proliferation, migration, apoptosis	proliferation (subcutaneous CDX model)	NR	NR	[57]
Nucleolin	HCC	promoter	migration	NR	NR	CCL20/CCR6 pathway	[58]
	HCC	promoter	proliferation, migration	NR	HDGF	PI3K/Akt pathway	[59]
	HCC	NR	NR	NR	C20orf204-189AA	NR	[60]
	GC	promoter	NR	NR	BTG2/SP1	TNF-α	[61]
	PDAC	promoter	migration	proliferation, migration, and angiogenesis (orthotopic CDA and CDX model)	NR	Endothelial cell activation, Ang-2 secretion	[62]
	PDAC	promoter	proliferation	proliferation (orthotopic CDX model)	NR	Autophagy via AMPK pathway	[63]
	CRC	promoter	NR	NR	VEGF/PI3K/AKT pathway	EMT pathway	[64]
	CRC	NR	NR	NR	P-Selectin Binding Protein	PI3K/p38/MAPK complex formation	[65]
	CRC	suppressor	proliferation	NR	LUCAT1	MYC	[66]
DAXX	GC	promoter	proliferation, migration	NR	RanBP2/RanGAP1	NR	[67]
	GC	suppressor	stemness, migration	NR	NR	HDAC-1/SNAIL3 axis	[68]
	CRC	suppressor	proliferation, migration	NR	NR	CD24	[69]
	CRC	suppressor	migration	NR	NR	ZEB1	[70]
	CRC	promoter	proliferation	proliferation (subcutaneous CDX model)	PI3KCA/AKT pathway	PI3KCA	[71]
	CRC	NR	NR	NR	NR	TCF4	[72]
ATRX	CRC	promoter	proliferation	NR	JMJD1A	NR	[73]
	PDAC	suppressor	NR	KRAS^G12D^ tumorigenicity (inducible knockout mouse model)	NR	NR	[74]
MCM2	HCC	promoter	proliferation	NR	miR-34a-5p	NR	[75]
	CRC	promoter	stemness	NR	miR-195-5p/497-5p	NR	[76]
TONSL	GC	suppressor	proliferation, migration	NR	TONSL-AS1	NR	[77]
RBAP48	GC	promoter	apoptosis, radiosensitivity	NR	NR	PI3K/AKT pathway	[78]
sNASP	HCC	promoter	proliferation	proliferation (subcutaneous CDX model)	NR	H3K9me1 modification	[79]
	GC	promoter	proliferation, apoptosis	NR	miR-29c	NR	[80]
IPO4	GC	promoter	proliferation, migration	NR	NR	NR	[81]
SPT2	CRC	promoter	proliferation	NR	NR	NR	[82]
SPT6	CRC	promoter	proliferation, migration, apoptosis, stemness, chemosensitivity	proliferation and migration (metastatic and subcutaneous CDX model)	NR	SND1/hTERT axis	[83]
ANP32E	PDAC	promoter	proliferation, migration	NR	NR	β-catenin	[84]
	PDAC	promoter	proliferation, stemness	proliferation (subcutaneous CDX model)	miR-202-5p	NR	[85]
	GC	promoter	proliferation, apoptosis	NR	NR	NR	[86]
VPS72	HCC	promoter	proliferation, migration	NR	NR	KAT5/PI3K/AKT pathway	[10]

Abbreviations: APE1: Apyrimidinic Endodeoxyribonuclease 1; ARF: ADP-ribosylation factor 1; ATF5: Activating Transcription Factor 5; CDK1: Cyclin-Dependent Kinase 1; BAX: BCL2-Associated X, Apoptosis Regulator; BTG2: BTG Anti-Proliferation Factor 2; CBX3: Chromobox 3; CCL20: C-C Motif Chemokine Ligand 20; CCR6: C-C Motif Chemokine Receptor 6; CDA: Cell line-derived allograft; CDX: Cell line-derived xenograft; DDX27: DEAD-Box Helicase 27; EC: esophageal cancer; EMT pathway: Epithelial–mesenchymal transition pathway; FBP1: Fructose-Bisphosphatase 1; hTERT: Human telomerase reverse transcriptase; KAT5: lysine acetyltransferase 5; KEAP1: Kelch-Like ECH-Associated Protein 1; LUCAT1: Lung Cancer-Associated Transcript 1; MMP9: Matrix Metallopeptidase 9; NR: not reported; NRF2: Nuclear factor erythroid 2-related factor 2; PDX: Patient-derived xenografts; PRDM8: PR/SET Domain 8; PRDX6: Peroxiredoxin 6; RASSF10: Ras Association Domain Family Member 10; RBAP48: RB Binding Protein 4, Chromatin Remodeling Factor; RNF2: Ring Finger Protein 2; SND1: Staphylococcal Nuclease and Tudor Domain Containing 1; SPHK1: Sphingosine Kinase 1; VPS72: Vacuolar Protein Sorting 72 Homolog.

## Data Availability

The present paper was registered in the OpenScience Framework registry (Registration DOI: https://doi.org/10.17605/OSF.IO/XE67H; accessed on 21 September 2022).

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
