# Peer review of "Histone Chaperones and Digestive Cancer: A Review of the Literature"

_cancers, 2022, doi:10.3390/cancers14225584_

Round 1
Reviewer 1 Report
I appreciate the authors' efforts to collect up-to-date information about the role of histone chaperones in digestive cancers. However, as a whole, the review lacks substance, it is not a comprehensive review, it is rather an extensive catalog of publications of histone chaperones mostly in digestive cancers. The mere listing of ten histone charepones and their implications in digestive cancers does not offer the reader any insight or critical information. Each chapter of a histone chaperone has a list of cellular processes, the main genes in digestive cancers they may be affected (without any substabtial discussion about causation or correlation), and how the chaperone is putatively associated with patients' prognosis. The authors do not present any studies on the functional analysis of these chaperones in digestive cancer mouse models, or any interpretation of how their overexpression may be linked to the formation or progression of digestive cancers, thus this lack of depth makes this review rather uninteresting to cancer biologists. As a cancer biologist and a mouse geneticist, I do not see value in this kind of review. Although there are no scientific inaccuracies throughout the manuscript the authors should rethink the scope of their review and need to focus either on a detailed example of how each of these chaperones is involved in digestive cancers or deconvolute genome-wide data meta-analysis focused on digestive cancers and identify how chaperones or their inhibitors may serve as therapeutic targets. Specific points :- During the identification process, it is more accurate to describe the removed records as overlapping between the 2 databases, rather than duplicated.
- There are 170 references listed at the end of the review, why do the authors present 80 articles in the "included" section of the flow diagram in Figure 1?
- The "screening process" for the literature search should go to the end or only in the supplementary section and the review should start with an overview of the role of different histone chaperones.
- A comprehensive figure for their role would be good. An excellent example of a similar review by a leader in the field is the following : DOI: 10.1016/j.gde.2022.101900
- In Chapter 4 "Shen et al. revealed that the FACT complex is regulated by NRF2 and could positively feedback regulate the transcription of NRF2..." this sentence does not make sense.
- In both tables, the word 'promotor' should be replaced with 'promoter'.
-
Identifying the authors with ORCID is highly recommended, especially when the corresponding author lists a generic email address.
Reviewer 2 Report
The review is well written and covers major histone chaperones and their observed relation to digestive cancer. However, the manuscript reads like a series of summary of different literature searches and regurgitation of the pubmed searches. For reviews, the authors will usually input some of their own personal interpretation of the results and critically looking at each histone chaperones' roles, and propose potential solution to resolve contradictory results. This review is lacking the above aspect a little.
Round 2
Reviewer 1 Report
I read the revised manuscript carefully and appreciate the effort the authors made to add more substance and accommodate my suggestions. I do find the revised version much better and I only have a few minor suggestions to keep the flow better. The first part of the results that the authors labeled 3.1 Search Results should be moved to the Literature Search part as well as Figure 2. As the authors stated they prefer to keep all this imformaiton in the main part of the review, still they should put their literarure search findings in one section. Part 3 should preferably be called 3. Histone Chaperones in Digestive Cancers or something similar and not 3. Results. Minor point, Figure 1 does not have letter A and B on the fIgure but authors use Fig.1A and Fig.1B in the text.
